# Quantitative Prediction of SYNTAX Score for Cardiovascular Artery Disease Patients via the Inverse Problem Algorithm Technique as Artificial Intelligence Assessment in Diagnostics

**DOI:** 10.3390/diagnostics12123180

**Published:** 2022-12-15

**Authors:** Meng-Chiung Lin, Vincent S. Tseng, Chih-Sheng Lin, Shao-Wen Chiu, Lung-Kwang Pan, Lung-Fa Pan

**Affiliations:** 1Department of Biological Science and Technology, National Yang Ming Chiao Tung University, Hsinchu 10041, Taiwan; 2Division of Gastroenterology, Department of Internal Medicine, Taichung Armed Forces General Hospital, Taichung 40044, Taiwan; 3Department of Computer Science, National Yang Ming Chiao Tung University, Hsinchu 10041, Taiwan; 4Department of Radiology, BenQ Medical Center, The Affiliated BenQ Hospital of the Nanjing Medical University, Nanjing 210000, China; 5Department of Pet Business Management, Taipei University of Marine Technology, Taipei 10001, Taiwan; 6Department of Medical Imaging and Radiological Science, Central Taiwan University of Science and Technology, Taichung 40044, Taiwan; 7Department of Cardiology, Taichung Armed Forces General Hospital, Taichung 40044, Taiwan

**Keywords:** inverse problem algorithm, SYNTAX, cardiovascular artery disease, computational analysis, artificial intelligence

## Abstract

The quantitative prediction of the SYNTAX score for cardiovascular artery disease patients using the inverse problem algorithm (IPA) technique in artificial intelligence was explored in this study. A 29-term semi-empirical formula was defined according to seven risk factors: (1) age, (2) mean arterial pressure, (3) body surface area, (4) pre-prandial blood glucose, (5) low-density-lipoprotein cholesterol, (6) Troponin I, and (7) C-reactive protein. Then, the formula was computed via the STATISTICA 7.0 program to obtain a compromised solution for a 405-patient dataset with a specific loss function [actual-predicted]^2^ as low as 3.177, whereas 0.0 implies a 100% match between the prediction and observation via “the lower, the better” principle. The IPA technique first created a data matrix [405 × 29] from the included patients’ data and then attempted to derive a compromised solution of the column matrix of 29-term coefficients [29 × 1]. The correlation coefficient, *r*^2^, of the regression line for the actual versus predicted SYNTAX score was 0.8958, showing a high coincidence among the dataset. The follow-up verification based on another 105 patients’ data from the same group also had a high correlation coefficient of *r*^2^ = 0.8304. Nevertheless, the verified group’s low derived average AT (agreement) (AT_avg_ = 0.308 ± 0.193) also revealed a slight deviation between the theoretical prediction from the STATISTICA 7.0 program and the grades assigned by clinical cardiologists or interventionists. The predicted SYNTAX scores were compared with earlier reported findings based on a single-factor statistical analysis or scanned images obtained by sonography or cardiac catheterization. Cardiologists can obtain the SYNTAX score from the semi-empirical formula for an instant referral before performing a cardiac examination.

## 1. Introduction

The quantitative prediction of the SYNTAX score for cardiovascular artery disease patients using the inverse problem algorithm technique as an artificial intelligence assessment in clinical diagnostics was evaluated in this study. The SYNTAX score is an angiographic tool to help cardiologists, interventionists, and surgeons to grade the complexity of coronary artery lesions. A higher SYNTAX score indicates a more complex condition and a worse prognosis in patients undergoing contemporary revascularization [1]. The symptoms of tiredness, wheezing, and swelling that often occur in clinical cardiovascular artery disease (CAD) are often mistaken for normal aging, so up to 90% of patients are unable to detect cardiovascular-artery-related symptoms at an early stage and often miss the golden treatment time [2]. Heart disease featured among the top ten causes of death worldwide from 2000 to 2019. More than 70,000 patients are hospitalized due to cardiovascular artery disease in Taiwan every year, according to WHO statistics [3]. Statistics from the Heart Failure Registration Program released in 2017 show that up to 32.3% of patients will be hospitalized again within six months, and the mortality rate within five years of being diagnosed with cardiovascular artery disease is nearly 50% [4]. Cardiovascular artery disease is a significant disease that should not be neglected. Moreover, patients suffering from cardiovascular artery disease usually deteriorate without proper treatment in advance.

Many researchers have noticed this crucial problem and tried to propose many preliminary predictions of the SYNTAX score to prevent cardiovascular artery disease in advance. For instance, Akboga et al. claimed that SYNTAX had a significant correlation with the ratio of monocytes to high-density lipoprotein cholesterol from the observation of 1229 patients [5]. Ikeda et al. found a significant correlation between carotid intima-media thickness and SYNTAX from 370 consecutive patients [6] and between carotid artery intima-media thickness and the plaque score from 501 consecutive patients [7]. Ikeda et al. also revealed that carotid artery ultrasound imaging and the ankle-brachial index could reasonably predict the severity of SYNTAX from 496 patient cases [8]. Rahmani et al. investigated the correlation between the Global Registry for Acute Coronary Events (GRACE) and SYNTAX for the risk stratification of 330 patients, although the regression correlation coefficient was as low as 0.116 [9]. However, properly grading the SYNTAX score in clinical diagnosis is quite problematic. As clearly depicted in Figure 1, two scenarios illustrate SYNTAX score grading from a cardiac X-ray examination. As shown, every lesion was graded according to its size or calcification with different weighted factors and eventually received scores of 47 (high) and 10 (low) in scenarios 1 and 2, respectively [1,2]. In contrast, seven essential factors (1. age; 2. mean arterial pressure; 3. body surface area; 4. pre-prandial blood glucose; 5. low-density-lipoprotein cholesterol; 6. Troponin I; and 7. C-reactive protein) were adopted as risk factors to satisfactorily predict the SYNTAX score using an inverse problem algorithm in this study. In doing so, 29 customized terms of a first-order nonlinear semi-empirical formula were derived via the STATISTICA 7.0 software to perform the analysis and provide reliable results on either numerical coincidence or clinical verification. A related discussion concerning the IPA technique or SYNTAX prediction is also included. A comparison of various forecasts of the SYNTAX score was also performed.

## 2. Methodology

### 2.1. Basics of the Inverse Problem Algorithm

In the first-order linear equation y=βx, *y* is the expected value, while the sensitivity of *x* to *y* is reflected by β. If *y* = *y* [405 × 1] is the expected value, also referred to as the actual SYNTAX score, which correlates with 29-term coefficients, *M* [29 × 1], then the respective correlation equation takes the following form:(1)Y=VM
(2)$$\left|\begin{array}{c} y_{1}\\ y_{2}\\ y_{3}\\ \vdots \\ y_{n} \end{array}\right|=\left|\begin{array}{ccccc} v_{11} & v_{12} & . & . & v_{1m}\\ v_{21} & v_{22} & . & . & v_{2m}\\ v_{31} & v_{32} & . & . & v_{3m}\\ \vdots & \vdots & \vdots & \vdots & \vdots \\ v_{n1} & v_{n2} & . & . & v_{nm} \end{array}\right|\left|\begin{array}{c} M_{1}\\ M_{2}\\ M_{3}\\ \vdots \\ M_{m} \end{array}\right|$$

If ∅ is the standard loss function, then
(3)$$\emptyset =\|VM-Y{\|}_{2}^{2}$$
(4)$$\nabla_{M}\emptyset =2\left(V^{T}\cdot VM-V^{T}Y\right)=0$$
(5)$$V^{T}\cdot VM=V^{T}Y$$
(6)$$M={\left(V^{T}\cdot V\right)}^{-1}\cdot V^{T}\cdot Y$$
where *V* and *V^T^* are the direct and transpose dataset matrices of the risk factors and cross-interactions between two factors [405 × 29]. For calculating the extreme values of the proposed function, according to *L’Hospital’s* rule, Equation (4) implies that the first-order total differential of the loss function Φ has a zero value. Then, the particular inverse matrix (*V^T^*·*V*) (cf. Equations (2)–(6)) is used to derive the column matrix of the 29-term coefficient *M* [10]. The computation is performed via the STATISTICA 7.0 default program, yielding a compromised solution with the minimal loss function Φ. This solution can be further customized according to user demand.

The IPA technique’s most available feature is that, besides providing a quantitative expectation of the particular syndrome based on several biological indices, it also forecasts the potential risk to medical staff when facing patients with no significant syndrome detected. In addition, solving the inverse matrix of biological datasets satisfies the convergence of numerical analysis. The derived semi-empirical formula offers an additional suggestion for clinical imaging diagnosis from any radiological facility, such as cardiac X-ray, sonography, or CT angiography.

### 2.2. The IPA Flowchart

The IPA technique in artificial intelligence can be schematized by the flowchart in Figure 2. It implies that the SYNTAX score in this study (as a quantified expectation value of the particular project) should be defined first. Next, one has to preset the number of risk factors that have to be orthogonal. Then, the estimated expectation value should be verified using data from another group of patients to ensure accuracy. Any failure in verifying or checking the program outcomes (loss function, variance, or correlation coefficient) via the STATISTICA 7.0 program requires going back to the preliminary stage to redefine the risk factors or increase the number of patients’ data. Otherwise, due to the limited data scope, the program may not converge to an acceptable range.

### 2.3. Semi-Empirical Formula Elaboration

In IPA, semi-empirical formulas contain only contributions from one factor and cross-interactions between two factors. Thus, all triple (*v*1 × *v*2 × *v*3, or *v*1 × *v*2 × *v*4, etc.) or quadruple (*v*1 × *v*2 × *v*3 × *v*4, or *v*1 × *v*2 × *v*3 × *v*5, etc.) cross-interactions among factors are ignored. In contrast, all multiple residual cross-interactions are merged into the final constant term as a minor oscillation to reach convergence of the numerical solution. The mathematical expression is defined as follows:


(7)
$$\begin{array}{l} v8=a1\times v1+a2\times v2+a3\times v3+a4\times v4+a5\times v5+a6\times v6+a7\times v7+a8\times v1\times v2\\ \;\;\;\;\;\;+a9\times v1\times v3+a10\times v1\times v4+a11\times v1\times v5+a12\times v1\times v6+a13\times v1\times v7\\ \;\;\;\;\;\;+a14\times v2\times v3+a15\times v2\times v4\;+a16\times v2\times v5+a17\times v2\times v6+a18\times v2\times v7\\ \;\;\;\;\;\;+a19\times v3\times v4+a20\times v3\times v5+a21\times v3\times v6+a22\times v3\times v7+a23\times v4\times v5\\ \;\;\;\;\;\;+a24\times v4\times v6+a25\times v4\times v7+a26\times v5\times v6+a27\times v5\times v7+a28\times v6\times v7\\ \;\;\;\;\;\;+a29 \end{array}$$


As depicted, the expectation value (*v*8, i.e., the SYNTAX score in this study) is always listed on the left side of the equation. In contrast, the right side contains the semi-empirical formula of seven variables (*v*1~*v*7).

### 2.4. SYNTAX Score and Seven Risk Factors

The SYNTAX score is the sum of the points assigned to each lesion identified in the coronary tree with >50% diameter narrowing in vessels above 1.5 mm in diameter. Further, the SYNTAX score is subdivided into three scenarios, namely, low (≤16), intermediate (16–22), and high (>22) [1]. In this study, the SYNTAX score was graded by seasoned cardiologists or interventionists when patients underwent cardiovascular examinations.

Seven essential biological indices were assigned as risk factors in this study: (1) age, (2) mean arterial pressure (MAP), (3) body surface area (BSA), (4) pre-prandial blood glucose (glucose AC), (5) low-density-lipoprotein cholesterol (LDL-C), (6) Troponin I (cTnI), and (7) C-reactive protein (CRP). MAP is a widely used parameter, reflecting the mean pressure in human arteries per complete cardiac cycle. It is considered a better indicator of perfusion to organs than systolic blood pressure (SBP) or diastolic blood pressure (DBP), being derived as follows: MAP = (SBP + 2·× DBP)/3. 3. The body surface area (BSA) strongly correlates with human metabolic mechanisms and is defined as $\sqrt{H\times W/3600}$) [m^2^] (*H*: height [cm]; *W*: weight [kg]). Glucose AC in fasting individuals is known to be maintained at a constant level at the expense of glycogen stores in the liver and skeletal muscle. LDL-C is one of the five major groups of lipoproteins that transport all fat molecules around the body in extracellular water. Troponin I (cTnI) is a cardiac and skeletal muscle protein that binds to actin in thin myofilaments and holds the actin–tropomyosin complex in place. The last factor, CRP, is an annular pentameric protein found in blood plasma, whose circulating concentrations rise in response to inflammation. It is an acute-phase protein of hepatic origin that increases following interleukin-6 secretion by macrophages and T-cells.

All risk factors should be normalized to the same domain range from −1 to +1 before executing the STATISTICA 7.0 program for the IPA structure’s incorporation of clinical data and the unification of each risk factor’s dimensionality. Each critical risk factor reading *X** is normalized via the following equation:(8)$$X^{*}=\frac{X-\frac{X_{max}+X_{min}}{2}}{\frac{X_{max}-X_{min}}{2}}$$
where *X*, *X_min_*, and *X_max_* are the respective risk factor’s original, minimum, and maximum readings (*V*_1_–*V*_7_). For example, for MAP (*V*_2_)’s maximum and minimum readings of 153 and 50 mmHg, respectively, the MAP values of case Nos. 100 or 183 were normalized from their original values (71 and 120) to the following ones: −0.6026 and +0.3550. Thus, the MAP scale range was normalized from −1.0 to +1.0.

The readings of the seven factors and their actual (original) SYNTAX scores before the normalization process (cf. Equation (8)) were obtained for 405 cardiovascular artery disease patients with their cardiac diagnoses reported in the Taichung Armed Forces General Hospital, Taiwan, from 1 January 2016 to 30 June 2021. In addition, another group of 105 patients with a similar syndrome was randomly assigned as a verified group from the original 555 (405 + 105 = 555)-patient group in the follow-up study. The survey was authorized by the Institutional Review Board (IRB) of the Tri-Service General Hospital, Taiwan (Permit No. B202005075). The individual results are given in Table 1.

### 2.5. Running STATISTICA 7.0 Program

The STATISTICA 7.0 default program [11] was used to execute the IPA algorithm. The correlation and cross-interaction among seven factors (cf. Equation (7)) were assigned and treated as nonlinear estimations, nonlinear models, and user-specified regressions with customized loss functions. The numerical simulations adopted the normalized data from 405 patients. The loss function was calculated explicitly via Rosenbrock and quasi-Newton numerical analyses, yielding the converged solution. Noteworthy is that alternative methods, such as Simplex, simples, or Rosenbrock pattern search, failed to obtain the minimum loss function that would satisfy user demands in this study.

The actual SYNTAX scores of cardiovascular artery disease patients were the expectation values of the computational results. Therefore, 11,745 individual data points (405 × 29 = 11,745) were included in the algorithm to optimize the compromised column matrix (405 × 1 = 405) of the SYNTAX scores of patients as a final numerical solution. In addition, twenty-nine terms containing one constant were incorporated into this algorithm to reveal any possible links among clinical factors. The loss function (Φ) was defined according to the total fluctuation between each theoretical and actual SYNTAX score for all 405 cardiovascular artery disease patients. The STATISTICA 7.0 program operation is visualized in Figure 3. One has to follow the proposed options and define the unique loss function to construct the coefficient matrix via the IPA.

## 3. Results

### 3.1. STATISTICA 7.0 Outcomes

Table 2 shows the precise data of the risk factors after normalization. As clearly illustrated, the mean value should approach 0.0 if the specific biological index of the patient group follows the normal distribution (range from −1.0 to +1.0). Accordingly, glucose AC (0.68), cTnI (0.90), and CRP (0.74) had high average values, whereas MAP (0.00), BSA (0.07), age (0.17), and even the SYNTAX score itself (0.11) fulfilled the definition of a standard normal distribution from a total of 405 patients’ statistical data.

### 3.2. Quantified Performance

Figure 4 depicts the calculated outcomes of STATISTICA 7.0. The customized loss function (Φ = [OBS−PRED]^2^) should be equal to zero in the case of a 100% match between practical observation and theoretical prediction, whereas this study derived a value of 3.177. The sample variance and regression correlation were 0.8958 and 0.9465, respectively, indicating the high coincidence of the derived prediction according to the original data matrix. The calculated coefficients of the 29-term semi-empirical formula, as defined in Equation (7), are listed in Table 3. Since all risk factors were normalized from −1 to +1, high coefficients corresponded to significant contributions in dominating the prediction of the SYNTAX score.

## 4. Discussion

### 4.1. Verifying the Predicted SYNTAX Score

Another group of 105 patients who were randomly adopted from the original patient group in this study was assigned as a verified group to verify the prediction of the SYNTAX score from the derived semi-empirical formula. In doing so, the biological indices of the verified group were input as a dataset matrix and then calculated to obtain the predicted SYNTAX score. Table 4 shows the detailed information of the verified group. As demonstrated, each risk factor’s maximum or minimum value also falls into a similar range to that of the original group. Figure 5 shows that the derivation of data from a verified group of 105 patients coincided with the original data from the group of 405 patients. As depicted, the two data groups consistently merged along the axis of the actual SYNTAX score. Specifically, the defined agreement (AT) equals [(actual − prediction)/actual] of the SYNTAX score. Therefore, the average AT_avg_ and standard deviation of the 105 ATs are 0.308 and 0.193, respectively, implying high agreement between the actual and predicted values of the SYNTAX score [12,13,14]. Figure 6 illustrates the distribution of 105 individual ATs in this study. As demonstrated, most ATs lie below 0.4, showing a reliable prediction of the SYNTAX score.

### 4.2. Dominant Factors of the SYNTAX Score Prediction

Either LDL-C (ranking: 2), age (3), BSA (5), glucose AC (6), or MAP (8) is the dominant risk factor in predicting the SYNTAX score, whereas cTnI (20) and CRP (27) are minor contributors according to the corresponding coefficient from the STATISTICA 7.0 program outcomes (cf. Table 3). Since all risk factors were normalized to eliminate their dimensionality in the preliminary stage and fit them to the interval between −1.0 and +1.0, the derived coefficient of any specific risk factor reflects its dominance in the semi-empirical formula. Although the individual factors in this study may not provide dominant contributions to the expectation value, their cross-interactions could strongly dominate the performance. According to IPA’s computational assumption, the cross-interaction between two factors (for instance, A (age) and B (cTnI) in this study) was interpreted as A × B and mathematically defined as a cross-product (A × B) with a vertical vector to both A and B. Thus, additional terms of cross-interactions between two factors in the semi-empirical formula provided alternative paths for optimizing the compromised solution. In addition, the assigned vector of either factor itself or the cross-interaction between factors created three specific degrees of freedom (DOFs) along the vector for optimizing the compromised solution in the numerical analysis.

### 4.3. Reducing the Number of Risk Factors

The 29-term semi-empirical formula based on seven risk factors could reasonably predict the SYNTAX score, according to the verification performed using another group of 105 patients. However, once the number of risk factors is decreased, the limited term of the correlated semi-empirical formula should also lose its high accuracy in the presumption of a robust designation. Accordingly, we reduced the number of risk factors in decreasing order of importance: LDL-C (7), age (6), BSA (5), glucose AC (4), MAP (3), cTnI (2), and CRP (1) (cf. Table 3). Thus, the semi-empirical formula corresponding to either six, five, or even only one factor could be defined via Equations (9)–(13). Restated, the number of risk factors decreased sequentially from the first (CRP) to the second and the last one (age) with a corresponding short-term semi-empirical formula. Thus, the last semi-empirical formula (*v*2) was defined according to LDL-C (*v*1) only since it provided the most dominant contribution to the prediction of the SYNTAX score (cf. Equation (14)).
(9)$$\begin{array}{l} v7=a1\times v1+a2\times v2+a3\times v3+a4\times v4+a5\times v5+a6\times v6+a7\times v1\times v2+a8\times v1\times v3\\ \;\;\;\;\;\;+a9\times v1\times v4+a10\times v1\times v5+a11\times v1\times v6+a12\times v2\times v3+a13\times v2\times v4\\ \;\;\;\;\;\;+a14\times v2\times v5+a15\times v2\times v6+a16\times v3\times v4+a17\times v3\times v5+a18\times v3\times v6\\ \;\;\;\;\;\;+a19\times v4\times v5+a20\times v4\times v6+a21v5\times v6+a22 \end{array}$$
(10)$$\begin{array}{l} v6=a1\times v1+a2\times v2+a3\times v3+a4\times v4+a5\times v5+a6\times v1\times v2+a7\times v1\times v3+a8\times v1\times v4\\ \;\;\;\;\;\;+a9\times v1\times v5+a10\times v2\times v3+a11\times v2\times v4+a12\times v2\times v5+a13\times v3\times v4\\ \;\;\;\;\;\;+a14\times v3\times v5+a15\times v4\times v5+a16 \end{array}$$
(11)v5=a1×v1+a2×v2+a3×v3+a4×v4+a5×v1×v2+a6×v1×v3+a7×v1×v4+a8×v2×v3+a9×v2×v4+a10×v3×v4+a11
(12)v4=a1×v1+a2×v2+a3×v3+a4v1×v2+a5×v1×v3+a6×v2×v3+a7
(13)v3=a1×v1+a2×v2+a3×v1×v2+a4
(14)v2=a1×v1+a2

The prediction from the STATISTICA 7.0 program based on various risk factors is reorganized in Table 5. As clearly illustrated, the regression curve reflects the prediction versus the actual SYNTAX score under various risk factors, and a good capability of the program prediction can be observed from either high sensitivity (i.e., the regression curve slope) or high coincidence (i.e., the correlation coefficient of the regression curve) [15]. In addition, the original prediction according to seven risk factors is also listed for comparison. As clearly illustrated, the accuracy of the SYNTAX score dropped with the reduced number of risk factors. Therefore, bountiful information on a patient’s biological index is always preferable for data collection in artificial intelligence.

In contrast, even the most dominant factor, namely, LDL-C in this study, cannot reasonably predict the SYNTAX core alone in reality since the respective correlation coefficient drops to a tiny 0.0644. However, with one additional factor added (age, as the second dominant factor), the coefficient increases to 0.2922, although it is still lower than the one derived from seven factors (0.8958) in this study.

### 4.4. Discussion of Similar Research Results Based on Various Risk Factors

Scholars have performed similar research using various risk factors to predict SYNTAX scores for cardiovascular artery disease patients from multiple perspectives. In particular, Akboga et al. [5] adopted the monocyte-to-HDL-C ratio to predict the SYNTAX score. The acquired clinical patient data were subjected to statistical analysis by SPSS, yielding linear correlations of the monocyte-to-HDL-C ratio with the SYNTAX score and C-reactive protein (with respective correlation coefficients of only 0.371 and 0.336). Ikeda et al. [7,8] applied the ultrasonography technique to measure the cardiovascular artery’s intima-media thickness or plaque, revealing a correlation between the SYNTAX score and the derived data. Alternatively, Rahmani et al. [9] recommended using the GRACE (Global Registry of Acute Coronary Events) score to predict the SYNTAX score. However, the regression results barely showed a positive correlation (*r*^2^ = 0.116). Studies by Kurtul et al. [16] and Sebastianki et al. [17] attempted to predict the SYNTAX score via the serum albumin concentration and the ankle-brachial index, respectively. However, these studies revealed that glucose AC, LDL-C, and creatine were also correlated, reflecting their coupled contribution to the SYNTAX score at a particular level. Noteworthy is that the single-factor regression technique is far beyond the complexity that multiple-factor correlation can reveal, and thus, a suitable IPA technique, as proposed in this study, can satisfactorily resolve the challenge in numerical analysis. In addition, Liu et al. [18] reported that systolic and diastolic echocardiographic parameters could also predict the SYNTAX score via the sonography technique but still needed more convincing results to demonstrate the true correlation, since the measured group of patients contained only 74 persons. The advantage of the IPA technique over those above is that it furnishes a reliable and rapid suggestion of the SYNTAX score for clinical cardiologists to instantly alert them to potential risks to patients before undergoing any solid examinations.

### 4.5. IPA Technique in Artificial Intelligence Applications

To further explore the application of the IPA technique in artificial intelligence, as described in this study, the prediction of the SYNTAX score using seven risk factors was portrayed by a color ladder diagram, as listed in Figure 7. In doing so, four out of seven risk factors were preset as average values to imply their general behavior in the CAD patient group because of their minor contributions to the accuracy of the SYNTAX score (cf. Table 3) [19,20]. The four minor factors were preset as MAP (0.0), glucose AC (0.68), cTnI (0.90), and CRP (0.74). Accordingly, the resulting readings after normalization were all equal to 0.0 for these risk factors (cf. Equation (8)). The three major factors, age (30–90 yr), BSA (1.0–2.2 m^2^), and LDL-C (10–300 mg/dL), were preset as the *X*-, *Y*-, and *Z*-axes, respectively, in this study. As clearly demonstrated, the SYNTAX score is high (>22) when LDL-C is higher than 100 mg/dL and becomes severe for high LDL-C (>250 mg/dL). Either young age or a small BSA is beneficial for maintaining a low SYNTAX score. In addition, cardiologists or interventionists can quickly obtain the suggested SYNTAX score by calculating the semi-empirical formula with only three major factors or obtain a precise outcome with all seven risk factors, as mentioned in this study.

## 5. Conclusions

The quantitative prediction of the SYNTAX score for cardiovascular artery disease patients using the IPA technique as an artificial intelligence assessment in clinical diagnostics was evaluated in this study. The 29-term semi-empirical formula was defined according to seven risk factors (age, mean arterial pressure, body surface area, pre-prandial blood glucose, low-density-lipoprotein cholesterol, Troponin I, and C-reactive protein). The correlation between actual and predicted SYNTAX scores reached *r*^2^ = 0.8958, implying that a highly coincident solution was obtained based on the dataset of 405 patients. The obtained formula was verified by a dataset of 105 patients with similar symptoms, yielding *r*^2^ = 0.8304. The derived average AT of the verified group (AT_avg_ = 0.308 ± 0.193) revealed a slight deviation between the theoretical prediction from the STATISTICA 7.0 program and the grades assigned by cardiologists or interventionists. The proposed IPA technique proved to be a valuable and reliable tool in helping clinical diagnosis. Patients can receive an instant SYNTAX score from personal biological indices before undergoing cardiovascular artery examination by either ultrasonography or cardiac catheterization.

## Figures and Tables

**Figure 1 diagnostics-12-03180-f001:**
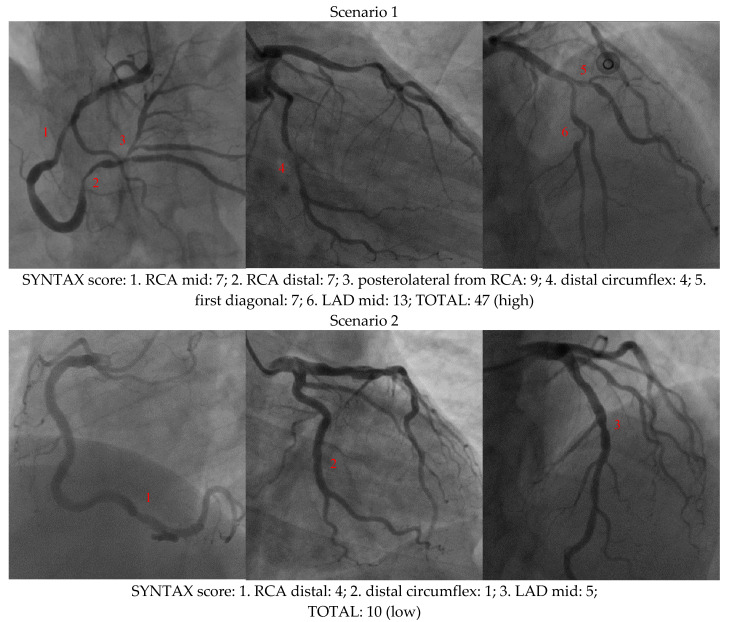
Two scenarios showing how the SYNTAX score was graded in a cardiac examination. The SYNTAX score was graded as 47 (**high**) and 10 (**low**) for scenarios 1 and 2, respectively.

**Figure 2 diagnostics-12-03180-f002:**
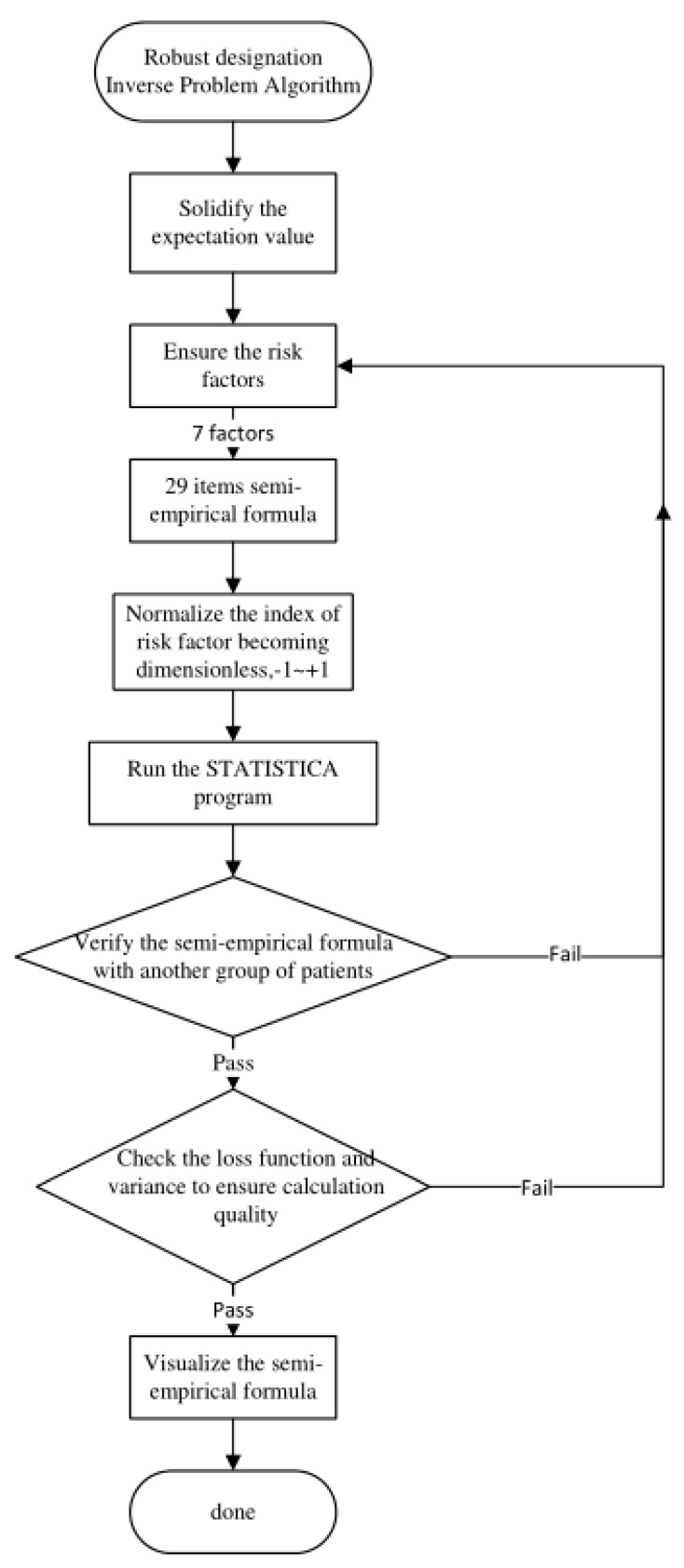
The flowchart of specific workloads illustrates how researchers apply the IPA technique in artificial intelligence.

**Figure 3 diagnostics-12-03180-f003:**
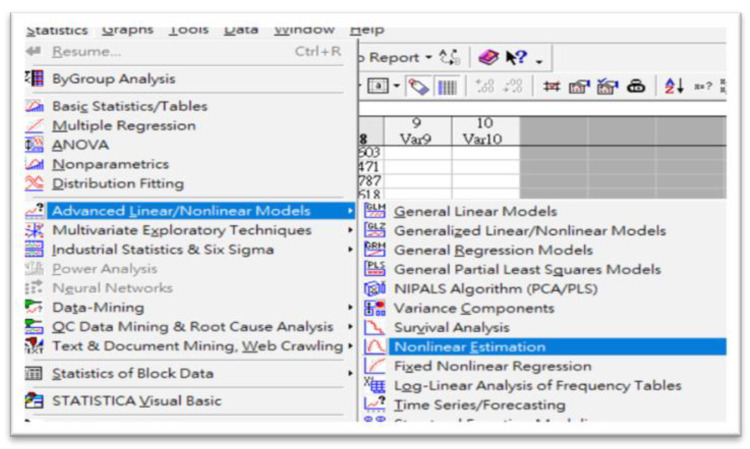
A typical STATISTICA 7.0 program in function. The user must follow the suggested options and define the unique loss function to obtain the coefficient matrix according to the IPA technique.

**Figure 4 diagnostics-12-03180-f004:**
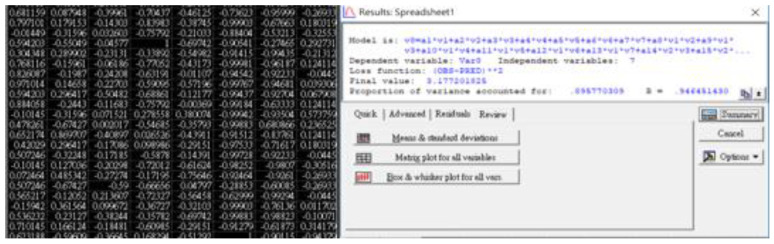
The calculated outcomes from STATISTICA 7.0 program. The customized loss function equals 0.0 if there is a 100% match between theoretical prediction and practical observation, whereas the derived value is 3.177 in this study.

**Figure 5 diagnostics-12-03180-f005:**
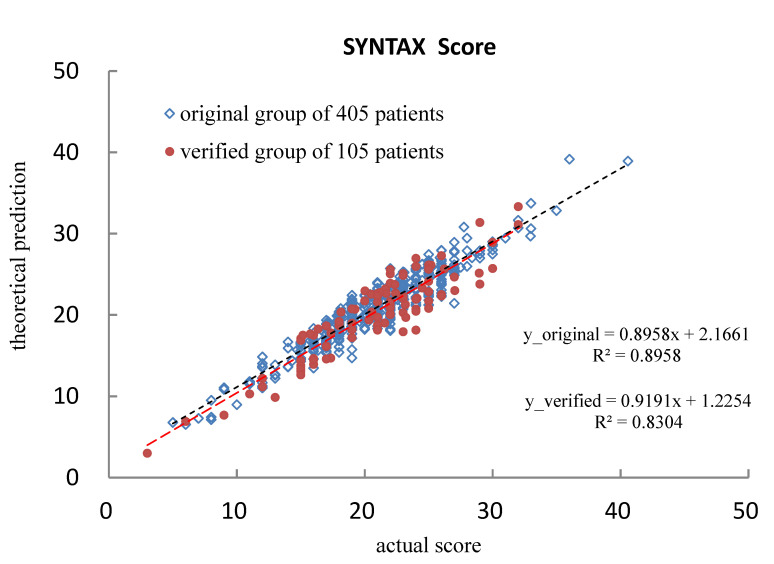
The actual and predicted SYNTAX scores for the original 405 patients and verified 105 patients, according to the STATISTICA 7.0-derived linear regression.

**Figure 6 diagnostics-12-03180-f006:**
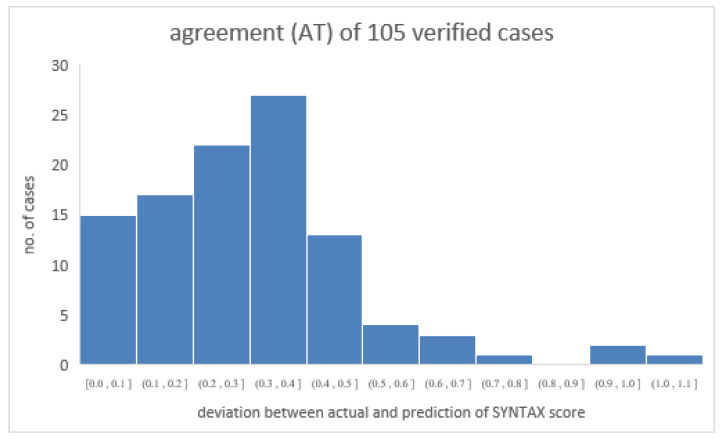
The distribution of 105 individual ATs in this study. As demonstrated, most ATs lie below 0.4, showing the convincing capability of the program to predict the SYNTAX score in reality.

**Figure 7 diagnostics-12-03180-f007:**
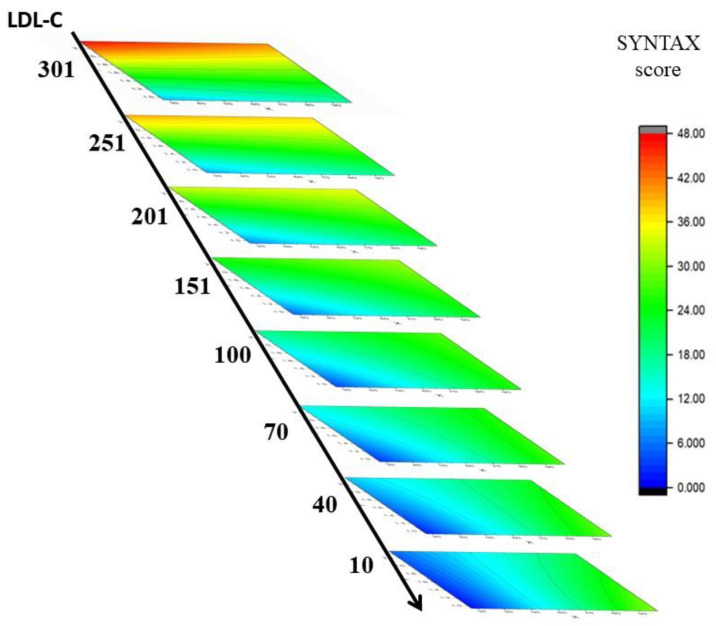
The major three factors, age (30–90 yr), BSA (1.0–2.2 m^2^), and LDL-C (10–300 mg/dL), were preset as *X*-, *Y*-, and *Z*-axes, respectively, in this study. As clearly demonstrated, the SYNTAX score is high (>22) when LDL-C is higher than 100 mg/dL and becomes severe for high LDL-C (>250 mg/dL).

**Table 1 diagnostics-12-03180-t001:** The readings of seven factors and actual (original) SYNTAX scores before the normalization process for 405 patients with cardiovascular artery diseases having their cardiac diagnoses reported in the Taichung Armed Forces General Hospital, Taiwan, from 2016 to 2021.

Factor	Range		Derived Data		
	Case No./Max.	Case No./Min.	Mean	Median	St. Dev
Age (yr)	75/98	71/29	70	71	14.1
MAP (mmHg)	386/153	276/50	101	101	18
BSA (m^2^)	264/2.40	63/1.09	1.70	1.70	0.20
Glucose AC (mg/dL)	15/689	272/54	156.6	126.5	87.4
LDL-C (mg/dL)	125/283	92/12	100.4	98.0	39.5
cTnI (ng/mL)	224/102.98	265/0.01	5.09	0.22	16.67
CRP (mg/dL)	193/42.50	368/0.01	5.51	2.47	7.19
SYNTAX score	191/41	267/5	20.8	21.0	4.9

**Table 2 diagnostics-12-03180-t002:** The readings of seven factors and the actual (original) SYNTAX scores after the normalization process. The mean value approaches 0.0 if the specific biological index of the patient group follows the normal distribution.

Factor	Range after Normalized		Derived Data after Normalized		
	Case No./Max.	Case No./Min.	Mean	Median	St. Dev
Age (yr)	75/+1	71/−1	0.17	0.22	0.41
MAP (mmHg)	386/+1	276/−1	0.00	0.00	0.36
BSA (m^2^)	264/+1	63/−1	0.07	0.07	0.31
Glucose AC (mg/dL)	15/+1	272/−1	0.68	0.77	0.28
LDL-C (mg/dL)	125/+1	92/−1	0.35	0.37	0.29
cTnI (ng/mL)	224/+1	265/−1	0.90	1.00	0.32
CRP (mg/dL)	193/+1	368/−1	0.74	0.88	0.34
SYNTAX score	191/+1	267/−1	0.11	0.10	1.00

**Table 3 diagnostics-12-03180-t003:** The coefficients of the 29-term semi-empirical formula (cf. Equation (7)) from the calculated outcomes of the STATISTICA 7.0 program. The factors were all normalized from −1 to +1. Thus, the large derived coefficients significantly dominate the performance of SYNTX score prediction.

Biological Index	Factor	Coefficient	After Normalization	
			Value	Rank
Age	A	a1	0.912696	**3**
MAP	B	a2	0.618889	8
BSA	C	a3	0.831829	**5**
Glucose AC	D	a4	0.825880	6
LDL-C	E	a5	1.175788	**2**
cTnI	F	a6	0.202330	20
CRP	G	a7	−0.103741	27
Age × MAP	A × B	a8	0.183358	21
Age × BSA	A × C	a9	−0.169841	22
Age × Glucose AC	A × D	a10	−0.304887	16
Age × LDL-C	A × E	a11	−0.295175	17
Age × cTnI	A × F	a12	−0.145316	24
Age × CRP	A × G	a13	1.252458	**1**
MAP × BSA	B × C	a14	−0.219970	18
MAP × Glucose AC	B × D	a15	0.376565	13
MAP × LDL-C	B × E	a16	0.757553	7
MAP × cTnI	B × F	a17	−0.135202	25
MAP × CRP	B × G	a18	0.398636	11
BSA × Glucose AC	C × D	a19	0.532427	9
BSA × LDL-C	C × E	a20	0.401664	10
BSA × cTnI	C × F	a21	0.124982	26
BSA × CRP	C × G	a22	0.021501	29
Glucose AC × LDL-C	D × E	a23	0.338031	15
Glucose AC × cTnI	D × F	a24	0.155761	23
Glucose AC × CRP	D × G	a25	0.202936	19
LDL-C × cTnI	E × F	a26	0.873143	**4**
LDL-C × CRP	E × G	a27	−0.347644	14
cTnI × CRP	F × G	a28	0.067604	28
Constant		a29	0.378574	12

**Table 4 diagnostics-12-03180-t004:** Detailed information of the verified group of 105 patients with similar cardiovascular artery diseases was randomly adopted from the same patient group as the original one.

Factor	Range		Derived Data		
	Case No./Max.	Case No./Min.	Mean	Median	St. Dev
Age (yr)	88/88	6/34	66	69	12.0
MAP (mmHg)	7/148	6/60	100	99	17.0
BSA (m^2^)	89/4.47	39/1.30	1.76	1.71	0.32
Glucose AC (mg/dL)	74/385	85/66	127.2	109.6	54.9
LDL-C (mg/dL)	84/187	95/45	104.9	99.9	41.5
cTnI (ng/mL)	74/102.98	1/0.01	3.69	0.09	14.74
CRP (mg/dL)	63/35.72	82/0.03	6.69	2.68	7.81
SYNTAX score	87/32	7/3	20.8	21.6	5.0

**Table 5 diagnostics-12-03180-t005:** Best-fitting parameters of the linear regression line. The results were calculated based on various numbers of risk factors via the STATISTICA 7.0 program. In addition, the original predictions according to seven risk factors are also listed for comparison.

Number of Factors	Number of Terms in the Regression Equation	Loss Function, Φ	Variance of Regression, s^2^	Linear Regression *y* = A*x* + B	Correlation Coefficient, *r*^2^
7	29	3.1772	0.8958	0.8958x + 2.1661	0.8958
6	22	9.6870	0.6822	0.6822x + 6.6042	0.6822
5	16	11.6247	0.6186	0.6185x + 7.9252	0.6186
4	11	16.1963	0.4687	0.4687x + 11.041	0.4687
3	7	20.1177	0.3400	0.3400x + 13.715	0.3400
2	4	21.5750	0.2922	0.2922x + 14.709	0.2922
1	2	28.5206	0.6437	0.0644x + 19.444	0.0644

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
