# Peer review of "Quantitative Prediction of SYNTAX Score for Cardiovascular Artery Disease Patients via the Inverse Problem Algorithm Technique as Artificial Intelligence Assessment in Diagnostics"

_diagnostics, 2022, doi:10.3390/diagnostics12123180_

Round 1
Reviewer 1 Report (Previous Reviewer 1)
Compared with the previous version, the manuscript is improved, and it can be accepted. All the previous comments were addressed.
Author Response
Thank you for the confirmation. To solidify our research findings and publish at highly ranked Journal is always our aiming in the research field indeed.
Reviewer 2 Report (Previous Reviewer 2)
See attached file

Round 2
Reviewer 2 Report (Previous Reviewer 2)
No further comment
This manuscript is a resubmission of an earlier submission. The following is a list of the peer review reports and author responses from that submission.
Round 1
Reviewer 1 Report
The authors presented methods for a quantitative prediction of SYNTAX score for cardiovascular artery disease patients using the Inverse Problem Algorithm (IPA) technique in preventive medicine. However, the language of the manuscript is very poor, and the related work is not correctly presented. Also, the images of the software presented were not relevant for the study. Although the results are interesting, the manuscript must be completely rewritten in a more scientific way.
Reviewer 2 Report
1. General comments:
When I read the abstract for examples
“The quantitative prediction of SYNTAX score for cardiovascular artery disease patients using the Inverse Problem Algorithm (IPA) technique in preventive medicine was explored in this study. The 29-term semi-empirical formula was defined according to seven risk factors: (1) age, (2) mean artery pressure, (3) body surface area, (4) pre-prandial blood glucose, (5) low-densitylipoprotein cholesterol, (6) Troponin I, and (7) C-reactive protein. Then, the formula was computed via the STATISTICA 7.0 program to conclude a compromised solution for 405 patients’ data set with a specific loss function [actual-predicted]2 as low as 3.177, whereas 0.0 implied a 100% match between each other via “the lower, the better” principle. The IPA technique first created a data matrix [405×29] from the included patients’ data and then attempted to derive a compromised solution of the column matrix of 29-term coefficients [29×1]. The correlation coefficient, r2, of the regression line for the actual verse predicted SYNTAX score was 0.8958 showing a high coincidence among the data set. The follow-up verification based on another 105 patients' data from the same group also had a high correlation coefficient, r2 =0.8304. Nevertheless, the verified group's low derived average AT (agreement) (ATavg= 0.308±0.193) also revealed a slight deviation between theoretical prediction from STATISTICA 7.0 program and graded by clinical cardiologists or interventionists. The predicted SYNTAX scores were compared with earlier reported findings based on the single-factor statistical analysis or scanned images from sonography or cardiac catheterization. Cardiologists can clue the SYNTAX score from the semi-empirical formula for instant referring before having any cardiac examination.”
I see nothing but a pure engineering work: a straightforward execution of a program. Research challenges are missing in this work
Again, the work is interesting but looks like an engineering work. The research challenges are not clearly stated.
2. Analysis:
Quality of the writing: structured coherently but pure design oriented
Abstract: Easy to understand
Introduction: well-written, context is clear to some extent.
Problematic: clearly-identified from an engineering point of view
Method: well-described
Application field: well-identified
Results: well-illustrated.
Conclusion: concise
Related works: more recent references are required, see recommendations below
Originality: The research originality is not clear. It looks more like a development work and programming
Recommendations:
Please define clearly the research challenge and the way to tackled them.
Round 2
Reviewer 1 Report
The manuscript is improved, and it can be accepted.
Author Response
thank you for your confirmation. we always try our best to solidify our research result into Journal article
Reviewer 2 Report
I recommended "Please define clearly the research challenge and the way to tackled them"
Authors replied "The reviewer's arguments for classifying this study as engineering work more related to programming than machine learning or image processing are very strong. However, we believe that our IPA-based methodology, which has been verified in several previous studies, e.g. [19, 20], can be easily incorporated into machine-learning systems (neural networks, etc.) operating with patient databases, becoming an integral part of computer-aided diagnostics. ..."
The fact that IPA-based methodology can be applied and that this methodology has been verified in several previous studies does not answer the question of research challenges.
